# Vegetation Dynamic Assessment by NDVI and Field Observations for Sustainability of China’s Wulagai River Basin

**DOI:** 10.3390/ijerph18052528

**Published:** 2021-03-04

**Authors:** Panpan Chen, Huamin Liu, Zongming Wang, Dehua Mao, Cunzhu Liang, Lu Wen, Zhiyong Li, Jinghui Zhang, Dongwei Liu, Yi Zhuo, Lixin Wang

**Affiliations:** 1College of Ecology and Environment, Inner Mongolia University, Hohhot 010021, China; chenpanpan@iga.ac.cn (P.C.); liuhmimu@aliyun.com (H.L.); bilcz@imu.edu.cn (C.L.); wenlu@imu.edu.cn (L.W.); zylee007@imu.edu.cn (Z.L.); jhzhang1001@126.com (J.Z.); liudw@imu.edu.cn (D.L.); zhuoyi@126.com (Y.Z.); 2Key Laboratory of Wetland Ecology and Environment, Northeast Institute of Geography and Agroecology, Chinese Academy of Sciences, Changchun 130102, China; zongmingwang@iga.ac.cn (Z.W.); maodehua@neigae.ac.cn (D.M.); 3College of Ecology and Environment, Collaborative Innovation Center for Grassland Ecological Security (Jointly Supported by the Ministry of Education of China and Inner Mongolia Autonomous Region), Hohhot 010021, China; 4Ministry of Education Key Laboratory of Ecology and Resource Use of the Mongolian Plateau, Hohhot 010021, China; 5National Earth System Science Data Center, Beijing 100101, China

**Keywords:** NDVI, vegetation dynamics, ecosystem communities, residual trend analysis, anthropogenic impacts

## Abstract

Accurate monitoring of grassland vegetation dynamics is essential for ecosystem restoration and the implementation of integrated management policies. A lack of information on vegetation changes in the Wulagai River Basin restricts regional development. Therefore, in this study, we integrated remote sensing, meteorological, and field plant community survey data in order to characterize vegetation and ecosystem changes from 1997 to 2018. The residual trend (RESTREND) method was utilized to detect vegetation changes caused by human factors, as well as to evaluate the impact of the management of pastures. Our results reveal that the normalized difference vegetation index (NDVI) of each examined ecosystem type showed an increasing trend, in which anthropogenic impact was the primary driving force of vegetation change. Our field survey confirmed that the meadow steppe ecosystem increased in species diversity and aboveground biomass; however, the typical steppe and riparian wet meadow ecosystems experienced species diversity and biomass degradation, therefore suggesting that an increase in NDVI may not directly reflect ecosystem improvement. Selecting an optimal indicator or indicator system is necessary in order to formulate reasonable grassland management policies for increasing the sustainability of grassland ecosystems.

## 1. Introduction

Agriculture and animal husbandry originate from grasslands; thus, human production and life are inseparable from grasslands [1]. Grassland dynamics play important roles in the ecological balance and in human economic development, and provide important ecosystem services [2,3]. Grassland vegetation features both seasonal and interannual change characteristics. Monitoring changes in vegetation using long-term remote sensing data can help to better understand and simulate the dynamic changes in terrestrial ecosystems and further reveal global vegetation dynamic trends and rules [4,5]. The normalized difference vegetation index (NDVI), obtained from the Global Inventory Modeling and Mapping Studies (GIMMS) Advanced Very-High Resolution Radiometer (AVHRR) or from the Moderate Resolution Imaging Spectroradiometer (MODIS), has been successfully used for global vegetation dynamic change monitoring. For example, Townshend and Justice used GIMMS AVHRR NDVI data from National Oceanographic and Atmospheric Administration (NOAA) 7 satellites to analyze the changes in vegetation dynamics, ecosystem types, and local rainfall in Niger, and concluded that high-resolution satellite data can be used to monitor vegetation [6]. Although the resolution of the AVHRR data is only 8 km, they were the dominant data from 1981 to 2006; therefore, they could not be replaced by other data [7]. MODIS NDVI data have significantly improved since 2000, but the lack of data before 2000 is a major disadvantage. GIMMS data and MODIS data have a similar dynamic range and greater time trend consistency [8,9,10]. Du et al. used the resampling method to compare and analyze GIMMS and MODIS NDVI data, fuse the two data sets, establish a consistent NDVI time-series, and monitor the vegetation dynamics in the Qinghai–Tibet Plateau [11]. However, the AVHRR data do not reflect the spatiotemporal changes of small- and medium-scale vegetation, whereas MODIS data do not conform to the needs of a study with a longer time-series.

It is important to solve the above mentioned problems by merging sensor data with different spatial, temporal, and spectral resolution in time and space, in order to generate remote sensing data with high spatial resolution and high temporal resolution characteristics. It is feasible to fuse GIMMS data with remote sensing data with a high spatial resolution to obtain a longer time-series data set. For example, Gao et al. proposed the spatial and temporal adaptive reflectance fusion model (STARFM), which used Landsat and MODIS imagery [12]. The model combined MODIS data and Landsat data to obtain surface reflectance data with high resolution in both time and space. On the basis of the STARFM algorithm, they proposed the spatial and temporal adaptive reflectance fusion model (STAVFM) for NDVI data with different spatial and temporal resolutions [13]. The algorithm is directly applied to the vegetation index and the time dimension weight is improved, according to the change in vegetation characteristics, such that an NDVI data set with high spatial and temporal resolution can be constructed. The enhanced spatial and temporal adaptive reflectance fusion model (ESTARFM) algorithm has been proposed to improve the algorithm on the basis of STARFM [14]. This algorithm fully takes into account the spatial position distance, spectral difference, and time difference between adjacent pixels and target pixels and, at the same time, enhances the accuracy of the simulated results in regions with large heterogeneity. The model inputs two sets of fine- and coarse-resolution images from the same or close time-series, successively, and then inputs the coarse-resolution images of the predicted target date. The generated result is fine-resolution images of the target date. In this study, the ESTARFM algorithm was used to construct an NDVI data set with high spatial and temporal resolution for the Wulagai River Basin. The reliability and validity of the reconstruction results were evaluated through a comparative analysis of the real and predicted values.

Several studies have shown that with the increase in population and grazing, grassland degradation is becoming an extremely important environmental problem in the grassland ecosystems of arid and semi-arid regions [15,16]. However, the main factors affecting grassland degradation have been controversial in academia [17,18,19]. GIMMS AVHRR NDVI and vegetation optical depth (VOD) data have been used to investigate the vegetation dynamics in the Mongolian Plateau from 1993 to 2012, and the vegetation dynamics caused by human factors were distinguished [20]. Grassland restoration policies play a key role in driving vegetation changes. A survey of vegetation changes in the Koshi River Basin of the Qinghai–Tibet Plateau and its response to climate change from 1982 to 2011 showed that climate change may play a crucial role in determining the trend of vegetation dynamics [21]. However, human contributions play a more prominent role in vegetation changes in some vulnerable ecosystems [22,23]. Studies on community structure have provided potential evidence regarding how climate or humans affect grassland vegetation changes [24,25]. Cislaghi et al. analyzed the relationships between the plant community and forage quality at seven different research sites in northern Italy. Their results showed that the succession of diverse plant communities is mainly caused by different local grazing intensities [26]. Similarly to human factors, competition-dominated communities are expected to be more stable and more resilient to climate change [27]. Studies have indicated that the relative importance of climate change and human activities on vegetation dynamics varies depending on the geographic location and ecosystem type [28,29].

The residual trend (RESTREND) method has been deemed to be an effective method for distinguishing the relative impacts of climatic factors and human activities on vegetation [30]. Previous studies have generally used precipitation and temperature data to build climate-based vegetation models, then derived the climate-based NDVI. Wessels et al. performed logarithmic calculations on precipitation data, analyzed the cumulative precipitation and its correlation with NDVI, established a regression model, and detected grassland degradation intensity using least squares regression [31]. Another study used the RESTREND method, combined with soil moisture and land degradation information, to analyze vegetation changes in sub-Saharan arid regions in West Africa [31]. Their results showed that the RESTREND method could explain land degradation and vegetation restoration in the arid regions of West Africa. However, studies that have analyzed vegetation dynamics based on the RESTREND method have only used NDVI data combined with climate data. Although some studies have used ground-based verification data (e.g., livestock density), it is still extremely difficult to verify whether the RESTREND method can represent the true succession status of plant communities [32]. The lack of field observation data has always been a challenge; therefore, in this paper, we utilized the RESTREND method and ground plot survey data to analyze the changes in vegetation dynamics and the characteristics of ecosystems in the Wulagai River Basin. Vegetation dynamics monitored by NDVI have good applications at a large scale. Does it accurately reflect vegetation conditions on a regional scale? Our results provide support for the sustainable development of the regional social economy and the implementation of grassland management policies.

In this study, we aimed to explore plant dynamics and the succession of ecosystem communities in the Wulagai River Basin, considering practical responses to the grassland management policies in the area over the past 22 years. The objectives of this study were: (1) To construct the NDVI time-series of the Wulagai River Basin from 1997 to 2018 by integrating GIMMS NDVI and MODIS NDVI data; and (2) to identify the driving force of vegetation dynamics in the Wulagai River Basin.

## 2. Materials and Methods

### 2.1. Study Area

The Wulagai River Basin is located in the northeast of Xilingol League, Inner Mongolia Autonomous Region, China, at longitudes from 117°25′ to 119°58′ E and latitudes from 44°19′ to 46°41′ N. The basin covers a total area of 27,362 km^2^ and its borders can be extracted based on the Wulagai River (Figure 1). The area falls within the criteria of a temperate continental monsoon climate; that is, it has an average annual temperature of −0.9 °C, average annual precipitation of 250–400 mm, and an altitude of about 850–1300 m [33]. The topography of the central part is mainly dominated by plains, but there are also low-lying hills, floodplains, terraces, and other landform types. According to the characteristics of the community structure (i.e., the composition of plant community species and biomass) in this area, we divided the whole area into three ecosystems (Appendix A
Figure A1). Specifically, the main ecosystem type is typical steppe, the northeast is meadow steppe, and the middle area is covered with riparian wet meadow. The species in this area are abundant, where the zonal vegetation mainly consists of *Stipa grandis*, *Stipa krylovii*, *Stipa baicalensis*, and *Leymus chinensis*, which are various types of forbs.

### 2.2. Data

#### 2.2.1. Normalized Difference Vegetation Index (NDVI) Data Sets

The GIMMS AVHRR NDVI data set was obtained from the Cold and Arid Regions Scientific Data Center of China (http://westdc.westgis.ac.cn) (accessed on 16 October 2020), with a spatial resolution of 8 km and an interval of 15 days from January 1997 to December 2006. The data set format was ENVI and the projection was Albers, with a total of 240 issues. The MODIS NDVI data set, with a spatial resolution of 250 m and an interval of 16 days from 2000 to 2018, was the MOD13Q1 data set downloaded from the USGS Land Processes Distributed Active Archive Center (http://ladsweb.nascom.nasa.gov/data/search.html) (accessed on 20 September 2020). The data set was in the Hierarchical Data Format (HDF) file-format, with a total of 434 issues. As a three-level grid data product with sinusoidal projection, MOD13Q1 is a corrected monthly data set, which contains corrections for sensor degradation, inter-sensor differences, cloud effects, solar zenith angle, and viewing angle.

#### 2.2.2. Plant Observations

According to the principle of random sampling, with consideration of the topography, terrain, altitude, spatial distribution of ecosystem types, hilly slopes, and sunny slopes in the watershed, fifty-five random sampling sites were determined in order to perform vegetation evaluation and collection in August 1997. We used GPS and other tools to conduct a point-by-point survey of the fifty-five sample points in July 2018 (Figure 2). GPS was used to record the latitude, longitude, and altitude of each sample point. A 10 × 10 m square at each sampling point was designed, and three squares (1 × 1 m) were then set on its diagonal [34]. The names, heights, and abundances of all species in each square were recorded. At the same time, each species was mowed and collected separately, and the fresh weight of all species was obtained on-site. The dry weight of each species was determined by taking each unit square from each sample point, according to the classified species, and oven-drying it at 75 °C for 24 h to a constant weight [34]. Each quadrant (1 × 1 m) was regarded as representative of a small-scale (area of 1 m^2^) community, whereas the three small-scale squares were combined into a large-scale community (with an area of 3 m^2^) at a sampling point. The aboveground biomass data of the sample points were the average values of the observations of this large-scale community.

#### 2.2.3. Meteorological Data

The meteorological data included monthly precipitation and temperature data from January 1997 to December 2018, which were obtained from the China Meteorological Data Network (http://data.cma.cn/) (accessed on 20 October 2020) and 13 meteorological stations around the study area. According to these data, we used the kriging method to generate precipitation and temperature distribution maps with the same resolution and geographic co-ordinate system as the NDVI data set, in order to study the correlation between climate factors and the spatial distribution of NDVI.

#### 2.2.4. Livestock and Crop Yield Statistics

We obtained livestock and crop yield data from the Statistics Bureau of the Wulagai management district. In this study, we assumed that livestock density and crop yield are direct anthropogenic drivers of vegetation dynamics in the area.

### 2.3. Methods

The following three steps were completed to achieve the goals of this study. First, a continuous NDVI data set from 1997 to 2018 was constructed. Secondly, the succession of plant communities in the ecosystems over the past 22 years was analyzed, using plant species diversity. Finally, the RESTREND method was used to distinguish the influence of human factors and natural factors on vegetation changes in the area, as well as to explore whether the NDVI can accurately reflect the community structure characteristics of different ecosystem community types.

#### 2.3.1. Building NDVI Data Sets

First, GIMMS and MODIS NDVI data were pre-processed. To enhance the quality of the data—especially to eliminate cloud pollution data and abnormal data—a Savitzky–Golay filter was used to smooth the NDVI data [35]. Then, the GIMMS and MODIS data were extracted, according to the watershed boundary of the study area, using the maximum value composite (MVC) method, in order to select the higher value of the half-month NDVI to obtain the monthly NDVI_max_ and, finally, the annual NDVI_max_ [35].

For consistency of data fusion, the GIMMS data were resampled to 250 m, in order to match the MODIS data. We used the ESTARFM fusion algorithm, and took the GIMMS NDVI annual maximum values from 1997 to 2005 and the MODIS NDVI annual maximum values from 2000 to 2005 as the basic input data, then obtained the predicted annual NDVI_max_ from 1997 to 2005 by calculation [13]. The input benchmark data for 2000–2005 were the adjacent data for those years. The forecast data and the corresponding period of MODIS NDVI data were evaluated for effectiveness. Consequently, the NDVI_max_ from 1997 to 2018 was successfully constructed. The specific process flowchart is described in Figure 3. *Tm* and *Tn* respectively represent basic input data, and *Tp* represents the data that need to be predicted.

Whether the high-spatial resolution data generated after fusion can accurately and truly compensate for actual missing data is directly related to the accuracy of the fusion data information extraction and the effectiveness of the subsequent research and utilization. Therefore, the evaluation and accurate analysis of simulated images are indispensable [36]. In order to test the prediction ability of the ESTARFM algorithm for NDVI derived from GIMMS and MODIS, as well as to check the consistency between the prediction results and the MODIS NDVI, we conducted a correlation analysis between the prediction results (ESTARFM NDVI) and the actual observation results (MODIS NDVI), and evaluated the reliability and effectiveness of the reconstruction results.

#### 2.3.2. The RESTREND Method

RESTREND is a pixel-based method that uses long-term time-series NDVI data for regression analysis. A regression relationship was established between the monitored NDVI_max_ and the accumulated precipitation and temperature data. Then, a statistical model was generated, which was used to calculate the predicted NDVI_max_ at each pixel [30]. Regression analysis was performed on the residuals between the observed and predicted NDVI_max_ of each pixel over time, in order to determine the direction, significance, and size of the change trend. A statistically significant downward trend of residuals indicates that vegetation degradation is mainly caused by human activities (such as overgrazing, land reclamation, and urbanization); whereas a significant increasing trend indicates that vegetation conditions have been improved. If the residuals show no trend over time, this is mainly due to vegetation changes caused by climatic factors [30,31].

As some previous studies have revealed, precipitation and temperature are the principal climate factors that regulate vegetation dynamics in arid and semi-arid areas [37,38]. However, different herbaceous biomes and ecosystem types respond differently to the variability of these two factors in different ways [39]. Therefore, to find the best correlation between different ecosystem types, we calculated the correlation for many different combinations of precipitation, temperature accumulation, and lag time for each NDVI pixel. The correlation between NDVI and precipitation has a time lag of about 1–12 weeks [40]. To calculate this interval and evaluate the true maximum correlation between NDVI and precipitation, a time lag of 0–3 months was used to calculate the correlation coefficient between NDVI and precipitation [41,42]. For the temperature variable, the accumulated temperature was selected for the regression calculation. The regression model with the highest R^2^ coefficient in the correlation was chosen to calculate the predicted NDVI values, residuals, and trends.

To better reflect the spatial variations in precipitation, temperature, and vegetation dynamics across the region, we used the entire regional dataset to conduct the regression between NDVI_max_ and meteorological factors and generate residuals. The annual NDVI_max_ was divided into six precipitation variables and two temperature variables at the pixel level, in order to establish correlations. The precipitation variables included cumulative precipitation with different time lags—cumulative precipitation from January to July, cumulative precipitation from April to July, cumulative precipitation from April to August, cumulative precipitation from June to August, and logarithmic forms of the abovementioned precipitation variables. Temperature variables included cumulative temperatures greater than zero degrees Celsius from January to August and their logarithmic forms. At each pixel, the regression equation with the highest R^2^ was expected to produce a residual. The statistical significance of the selected regression equation was tested, and only variables with significant regression models (*p* < 0.05) were applied in subsequent analyses.

Then, based on the two-sided t-distribution, combined with three confidence intervals (0.05, 0.10, and 0.25), eight categories were defined to test the trend and significance of the residuals. The details are as follows, and the decreasing trend (negative slope) was divided into four categories: D1 (|t statistics| > t_0_._05_ (n-2)), D2 (|t statistics| > t_0_._10_ (n-2)), D3 (|t statistics| > t_0_._25_ (n-2)), and DNC (|t statistics| < t_0_._25_ (n-2)). Similarly, there were four levels of increasing trend (positive slope), as follows: I1 (t statistics > t_0_._05_ (n-2)), I2 (t statistics > t_0_._10_ (n-2)), I3 (t statistics > t_0_._25_ (n-2)), and INC (t statistics < t_0_._25_ (n-2)). That is, the *p*-value of the trends represented by D1, D2, and D3 (and I1, I2, and I3) were 0.05, 0.10, and 0.25, respectively; DNC and INC denote visually discernable decreasing and increasing trends that were not statistically significant at *p*-values of smaller than 0.25.

## 3. Results

### 3.1. NDVI Dynamic Change Characteristics

We completed a comparative analysis and correlation scatter plot of the prediction results and observation results. Figure 4 shows the comparative analysis of local features from 1997 to 2005. From a visual point of view, there was a definite difference between the predicted NDVI and the MODIS NDVI. Details of the predicted NDVI in some areas were not as good as the MODIS NDVI, but the spatial information of the two was basically the same. As MODIS data began in 2000, our correlation scatter plot ranges from 2000 to 2005 (Figure 5). In the entire area, ESTARFM predicted that the total R^2^ of NDVI and MODIS NDVI was between 0.65 and 0.78, whereas the fitted RSME was between 0.0040 and 0.0065. These results show that the predicted NDVI and MODIS NDVI had good consistency and that the ESTARFM algorithm had a high predictive ability for NDVI in the Wulagai River Basin. Consequently, it was concluded that this method can effectively reconstruct the MODIS NDVI time-series.

It can be seen, based on the NDVI values of the three ecosystem types, that the meadow steppe community was intuitively the highest (i.e., between 0.57 and 0.80; see Figure 6). The NDVI values of the riparian wet meadow and the typical steppe were between 0.40 and 0.63 and between 0.44 and 0.69, respectively (Figure 6). It can be concluded that the changes on the time scale were all increasing. In addition, the NDVI of the riparian wet meadow had the fastest rising trend, the meadow steppe was second, and the typical steppe change was the slowest; however, it still showed a general upward trend. We also analyzed the average annual precipitation at the Wulagai weather station and found that the fluctuation tendency of NDVI was almost the same, which also echoed the time lag effect of precipitation (Figure 6).

### 3.2. Correlation between NDVI_max_ Spatial Distribution and Climatic Factors

In order to analyze the spatial distribution pattern of the relationship between NDVI_max_ and climate, we analyzed the pixel-level regression between cumulative precipitation, cumulative temperature, and NDVI_max_ with different time lags between 1997 and 2018 for three ecosystem types. The areas of the three types of ecosystems were different and we therefore selected the number of random pixels according to their proportion in the total study area. For the meadow steppe, typical steppe, and riparian wet meadow, we selected 200, 500, and 40 random pixels, respectively. The results demonstrated that, for the three ecosystem types, the best cumulative rainfall times were from April to July for 1997 to 2018 (Table 1).

The NDVI_max_ and precipitation were significantly related to 43.73% of the pixels of the meadow steppe, 51.65% of the pixels of the typical steppe, and 42.60% of the pixels of the riparian wet meadow (Table 1). The R^2^ relationships for the pixel-level regression are shown in Figure 7.

Compared with precipitation, the intensity and spatial extent of the effect of temperature on NDVI_max_ were relatively weak. The results showed that NDVI_max_ was negatively correlated with temperature in the study area, and the correlation between the three ecosystem types and the two temperature variables was not statistically significant (Table 1). This is in agreement with the results of another study, in which the effect of temperature change on NDVI_max_ was small [34]. The climate factors related to NDVI_max_ mainly included the precipitation time-series; thus, removing the precipitation signal is equivalent to eliminating the climatic impact on the NDVI time-series. Consequently, we only selected precipitation data and NDVI_max_ for the linear regression model and analyzed the trend of residual change.

### 3.3. Residual Analysis

Over the entire study period, among the residuals of NDVI_max_ and precipitation regression, the residuals of most pixels of each ecosystem type did not show statistical significance. Therefore, we divided the period of 22 years into three sub-periods according to changes in land-use policies (i.e., 1997–2003, 2003–2007, and 2008–2018), which represented the transition from free grazing to the beginning of the pasture management policy, the beginning of pasture contracting and basic pasture delineation phases, and the implementation of pasture restoration policies after pasture contracting, respectively. The residuals of the three sub-periods showed different trends. We determined the trends of the residuals using the slope of the regression line and the statistical significance (Figure 8 and Table 2).

The percentage of pixels in this area that showed a decreasing trend (D) reached its highest (36.3%) in the sub-period 1997–2003 and lowest in the sub-period 2003–2007 (1.8%). The percentage of pixels exhibiting an increasing trend (I) was highest in the sub-period 2008–2018 (27%), whereas those in the other two sub-periods were lower; that is, 1997–2003 (4.9%) and 2003–2007 (23.3%). In the first sub-period, the percentage of pixels that showed a significant but statistically insignificant decreasing trend (DNC) was 37.8%, whereas the percentages of pixels in the latter two sub-periods were similar, both below 20% and showing statistical significance (Table 2). However, the percentage of pixels that showed an apparent, but statistically insignificant, increasing trend (INC) had the highest percentage in the second sub-period (59%), while there was a large difference between the other two sub-periods (21.1% and 39.1%; see Table 2).

The residuals of each ecosystem type showed different trends for the three sub-periods (Figure 8). Most of the pixels of each of the three ecosystem types (about 59% in total) showed DNC or INC patterns, whereas 41% of the pixels showed an increasing or decreasing trend, indicating that there was no consistent trend during the sub-period 1997–2003 (Figure 9 and Table 2). However, during the sub-period 2003–2007, more pixels showed an upward residual trend, with a significant increase during the sub-period 2008–2018 (Figure 9 and Table 2).

There was a significant increase in livestock density and crop yield in the Wulagai management area from 1997 to 2018. The number of livestock in the management area increased, then decreased (i.e., from 300,000 head to 877,000 head and finally to 580,000 head; see Figure 10). The total crop yield fluctuated significantly in individual years and the overall trend continued to increase (i.e., from 3800 tons to 63,566 tons; see Figure 10). From 1997 to 2000, the trend of the residual error was not statistically related to the rate of change in livestock density or the total crop yield during the period (Table 3). From 2001 to 2018, the trend of the residual error and the rate of change in livestock density were significantly negatively correlated (Figure 11a and Table 3), whereas there was a significant positive correlation with the total crop yield (Figure 11b and Table 3).

### 3.4. Community Structure Characteristics of Ecosystem Types in the Wulagai River Basin

The study area is mainly dominated by Gramineae, Compositae, Leguminosae, and Rosaceae. We found that a total of 32 families, 96 genera, and 156 species were monitored in 1997, which had increased to 32 families, 103 genera, and 184 species in 2018.

In the meadow steppe ecosystem, the plant community type was a *S. baicalensiss* + *Filifolium sibircum* community in 1997, and a *S. baicalensiss* + *Carex korshinskyi* community in 2018. Moreover, the average plant height and aboveground biomass of the ecosystem in 2018 were higher than in 1997. Regarding the species composition, in 2018, the proportion of *F. sibircum*—which occupied as the dominant species in 1997—decreased (Table 4). In 1997, the typical steppe ecosystem was a *S. grandis* + *L. chinensis* community, which succeeded to a *S. grandis* + *S. krylovii* + *L. chinensis* community in 2018. The dominant species in the community changed from *S. krylovii* to *S. grandis*, the average plant height and aboveground biomass decreased, and the proportion of *Euphorbia fischeriana* and *Artemisia frigida* in species composition increased from 1997 to 2018 (Table 4). In the riparian wet meadow ecosystem, the *Agrostis alba* + *Potentilla anserina* community in 1997 became a *C. korshinskyi* + *Hemerocallis minor* community in 2018, and the main species composition also changed. In 2018, the aboveground biomass was 233.16 ± 2 g/m^2^ and the average plant height was 18.78 ± 0.3 cm, which was higher than that in 1997 (Table 4), but the number of species in the plot and the single species occupying the dominant position had not changed.

## 4. Discussion

### 4.1. The Dynamic Changes and Influencing Factors of NDVI in Different Ecosystems

The time-series changes in NDVI of different ecosystems in the Wulagai River Basin are shown in Figure 6. The lowest point of the NDVI trough appeared in 2007, which is consistent with the pasture management policy of the Wulagai Management Area. The government completed the confirmation of the contractual rights of the pasture from 2004 to 2006, and it began to be managed by various herders in 2007. There have been significant fluctuations in the NDVI due to policy changes. The annual precipitation in the Wulagai River Basin has also changed, corresponding with the fluctuating NDVI (Figure 6); for example, there was less precipitation in 2006 and 2007, and the environment was relatively dry. The combined action of natural factors and human factors caused the NDVI to experience greater fluctuations. Several previous studies which have analyzed vegetation trends based on remote sensing data have shown that the net primary production of the Xilingol grassland increased from the early 1990s to the early 2000s [43,44]. These results indirectly indicate that land-use policies and pasture management practices during this period had a positive impact on grasslands.

Affected by the atmospheric circulation in Southeast Asia, Inner Mongolia entered a flood season from west to east. The flood situation in May 1998 was very severe, and the precipitation in the study area reached its peak at this time. The NDVI in 1998 was slightly lower than that in 1997, caused by the flooding of grasslands and poor vegetation conditions (Figure 6). On the basis of the analysis, the remaining trends and the impact of precipitation effects were assessed; specifically, the performance of grassland vegetation changes in the Wulagai River Basin from 1997 to 2003 was manifested by a decline in vegetation coverage and biomass, which have gradually improved since 2004 (Figure 6 and Figure 8, Table 4). The grassland degradation in Inner Mongolia, based on field survey data, has been reported by Jiang et al. [45]. However, the vegetation in the Xilingol League showed a tendency to deteriorate in the late 1990s [46]. Therefore, correct identification of climate change and vegetation dynamics caused by human activities is essential for the implementation of appropriate land-use policies and grassland management.

### 4.2. The Influence of Anthropogenic Factors on the Residual Trend

In this study, the declining trend of the residuals of the three ecosystem types was mainly due to vegetation degradation from 1997 to 2003. The residuals of the typical steppe and riparian wet meadow ecosystems showed *p*-values < 0.1, whereas from 2003 to 2007 the upward trend indicated that the vegetation had improved. The residuals increased significantly, indicating the significant improvement of vegetation from 2008 to 2018. Furthermore, the same trend in the same period has also been proven in other studies [46,47]. From 1993–2001, both the number of livestock and the output of crops increased slightly (Figure 10). However, their correlations with NDVI residuals were not statistically significant (Table 3). This result may have been affected by the extreme drought in the area in 1997, the floods in 1998, and the ecological restoration plan that began in 2000 [28]. Specific measures to restore degraded grassland ecosystems have included the control of livestock numbers and grazing in specific areas or seasons [48]. The increase in surplus trend was closely related to the grazing prohibition policy, whereas the decrease in the remaining trend was related to an increase in livestock density (Figure 11a). Our field observations indicated that overgrazing had significantly reduced vegetation cover and aboveground biomass (Figure 6 and Table 4) and, combined with pasture management policies at that time, we concluded that grazing was the main driving force of grassland vegetation changes in the Wulagai River Basin from 1997 to 2003 [33]. The grassland in the Wulagai management area gradually changed from collective management to contractual confirmation of rights of pastoralists from 2003 to 2007, which was a significant change of policy during the research period. Consequently, the residual direction changed, as compared with that at the end of the 1990s. The drought conditions during the study period (2006–2007) only changed the residual rate and did not cause a change in direction. This also indicates that the influence of climatic factors on residuals is smaller than that of human factors [49,50], thus highlighting that the correct identification of climate change and vegetation changes caused by human activities is critical to the implementation of appropriate land-use policies and grassland management.

### 4.3. Comparison of NDVI and Community Structure Changes in the Same Ecosystem

The slope of NDVI for each ecosystem type was distinct, and they all showed an upward trend (Figure 6). In addition, the overall NDVI also showed an upward trend. The change in precipitation led to a corresponding change in vegetation fluctuation, in agreement with Zhou et al., who also reported that the NDVI showed an upward trend when rainfall increased [51]. A comparison of the results of two field observations (in 1997 and 2018) showed that both the biomass and plant height of the meadow steppe ecosystem increased, indicating a positive succession trend. Likewise, the NDVI results showed a similar trend. The NDVI of the typical steppe was the smallest among the three ecosystem types, but it still showed an overall upward trend, which was inconsistent with the field observation results. In the riparian wet meadow ecosystem, field observations showed a trend of degradation, whereas the NDVI results showed an increasing trend. Similarly, studies have shown that the results from field observation measurement data of Inner Mongolian grasslands differ from those derived from NDVI remote sensing data [45], indicating that it cannot completely represent the true growth status of the plant ecosystem. Some annual plants have replaced the dominant species in this ecosystem, showing higher NDVI values in remote sensing imagery; however, they are representative of grassland degradation [52]. The growth status of different ecosystems corresponds to different NDVI trends, such that an increase in NDVI trend alone cannot fully characterize the improvement of plant communities in the regional ecosystem. Therefore, focusing only on the results of remote sensing vegetation data, without considering the impact of climate and human factors, may lead to the incorrect implementation of pasture management and land-use policies.

## 5. Conclusions

Integrating GIMMS NDVI data, MODIS NDVI data, and the field investigation of plant communities, combined with the residual trend method, we analyzed the vegetation dynamics and ecosystem community structure characteristics of the Wulagai River Basin, distinguishing vegetation changes caused by climatic and human factors. Our study revealed that NDVI values in the Wulagai River Basin showed an overall gentle upward trend. The residual analysis indicated that human factors were the main driving forces of the vegetation dynamics, and that the implementation of the new pasture management policy should help to curb the increase in stocking rates and therefore increase the vegetation area. However, the field observation results were not completely consistent with the NDVI results. During the study period, the species distribution of the meadow steppe ecosystem was relatively even and the ecosystem was stable. Additionally, the typical steppe and riparian wet meadow ecosystems experienced declines in productivity and the replacement of dominant species, such that the ecosystems can be considered relatively fragile. Annual vegetation in these communities has replaced the original dominant species, which has led to an increase in the NDVI, thus masking the degradation trend of the grassland. On a regional scale, the NDVI results alone cannot explain the improvement or deterioration of all grassland types, nor can it reflect the changes in the dominant species of plant communities. Therefore, when using the NDVI to monitor grassland conditions, it needs to be verified through field observation data. This research provides important scientific information for formulating local grassland protection and restoration policies.

## Figures and Tables

**Figure 1 ijerph-18-02528-f001:**
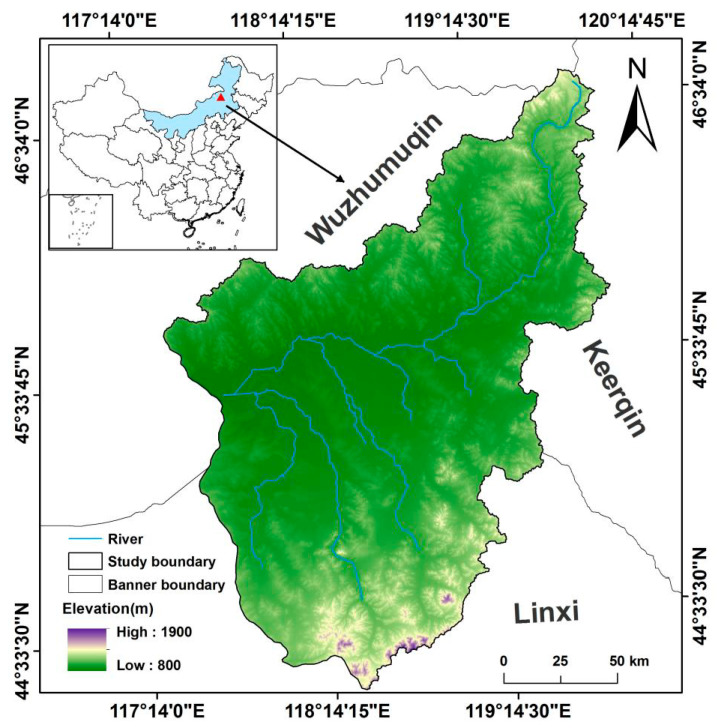
Location of the Wulagai River Basin and the digital elevation model (DEM).

**Figure 2 ijerph-18-02528-f002:**
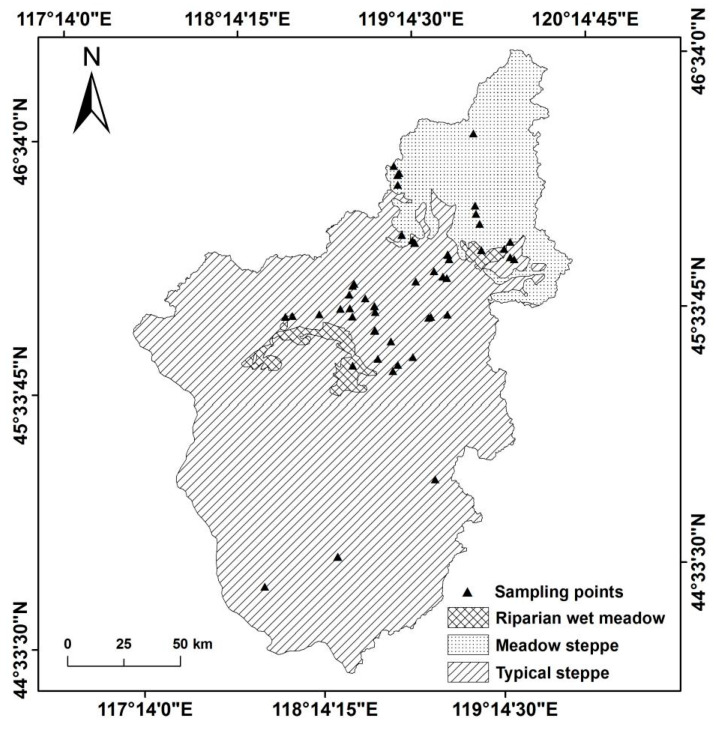
Spatial distribution of the surveyed plots and different ecosystem types in the Wulagai River Basin.

**Figure 3 ijerph-18-02528-f003:**
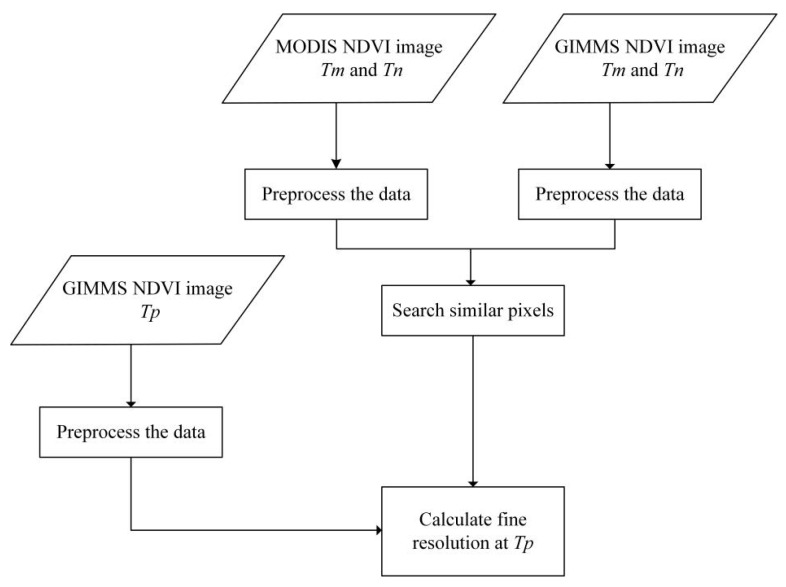
The flowchart of the enhanced spatial and temporal adaptive reflectance fusion model (ESTARFM) algorithm. GIMMS, Global Inventory Modeling and Mapping Studies; MODIS, Moderate Resolution Imaging Spectroradiometer; NDVI, normalized difference vegetation index. *Tm* and *Tn* respectively represent basic input data, and *Tp* represents the data that need to be predicted.

**Figure 4 ijerph-18-02528-f004:**
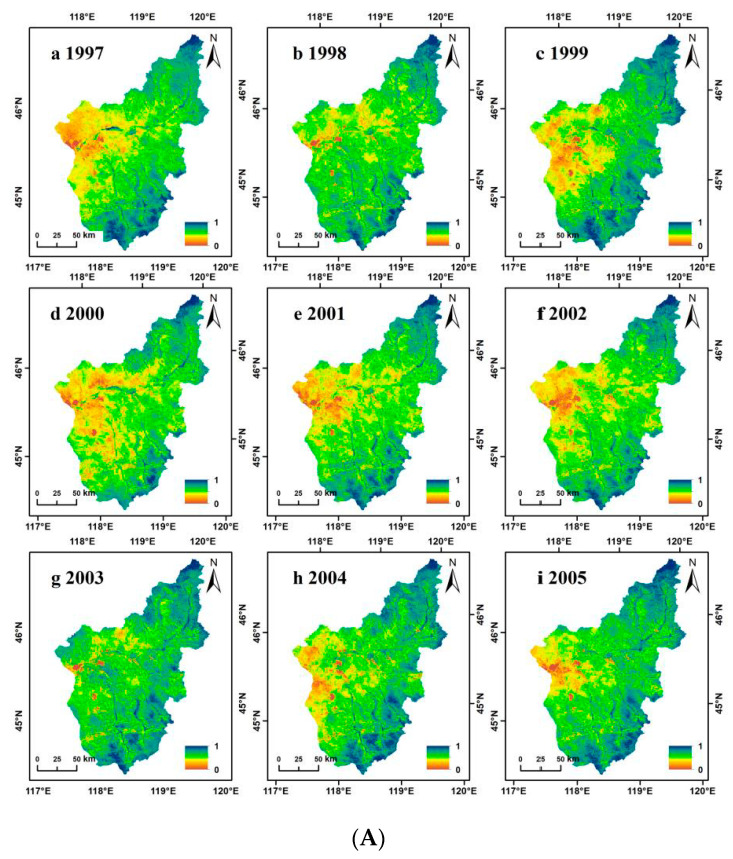
(**A**) Prediction images (enhanced spatial and temporal adaptive reflectance fusion model normalized difference vegetation index; ESTARFM NDVI) from 1997 to 2005. (**B**) Actual images (moderate resolution imaging spectroradiometer normalized difference vegetation index; MODIS NDVI) from 2000 to 2005.

**Figure 5 ijerph-18-02528-f005:**
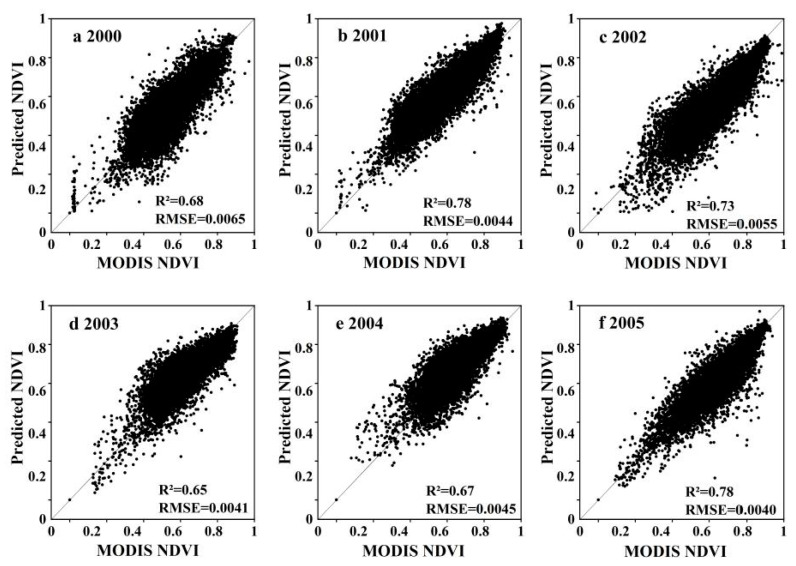
Linear regression between the ESTARFM-predicted NDVI and MODIS NDVI in the entire study area. The *p*-values of hypothesis tests in each year are all < 0.01.

**Figure 6 ijerph-18-02528-f006:**
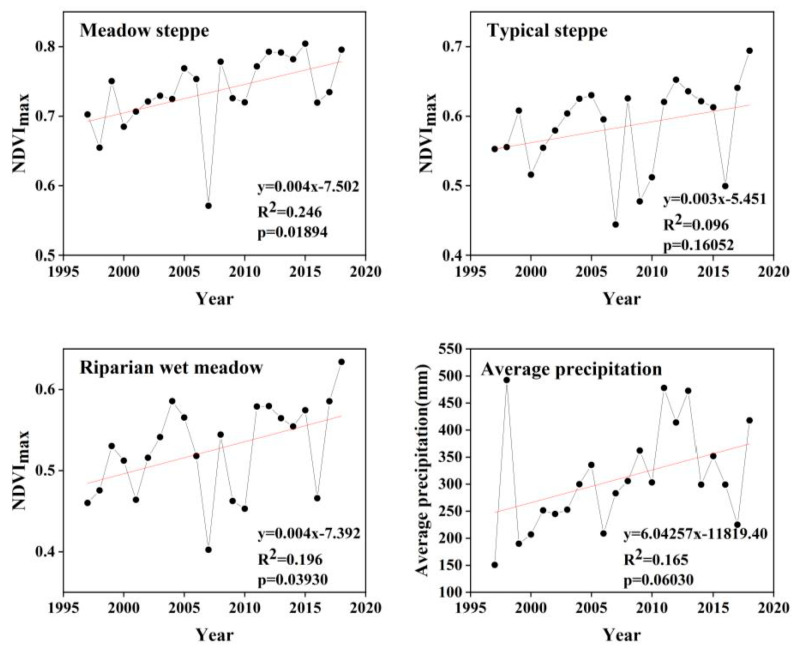
NDVI_max_ and precipitation trends for the three ecosystem types from 1997 to 2018.

**Figure 7 ijerph-18-02528-f007:**
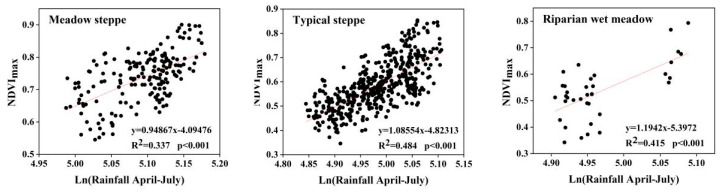
Correlation between NDVI_max_ and precipitation from April to July for different ecosystem types.

**Figure 8 ijerph-18-02528-f008:**
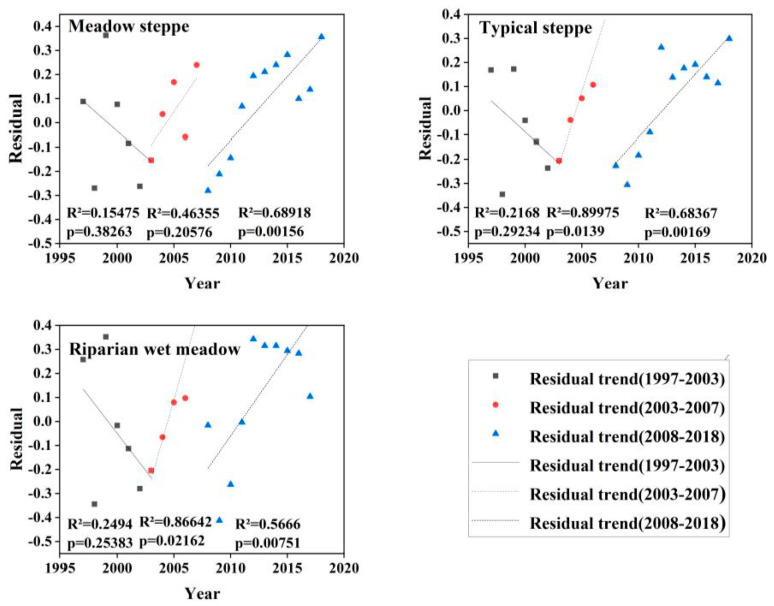
Residual trend graphs of different types of ecosystems for three time-series: 1997–2003, 2003–2007, and 2008–2018.

**Figure 9 ijerph-18-02528-f009:**
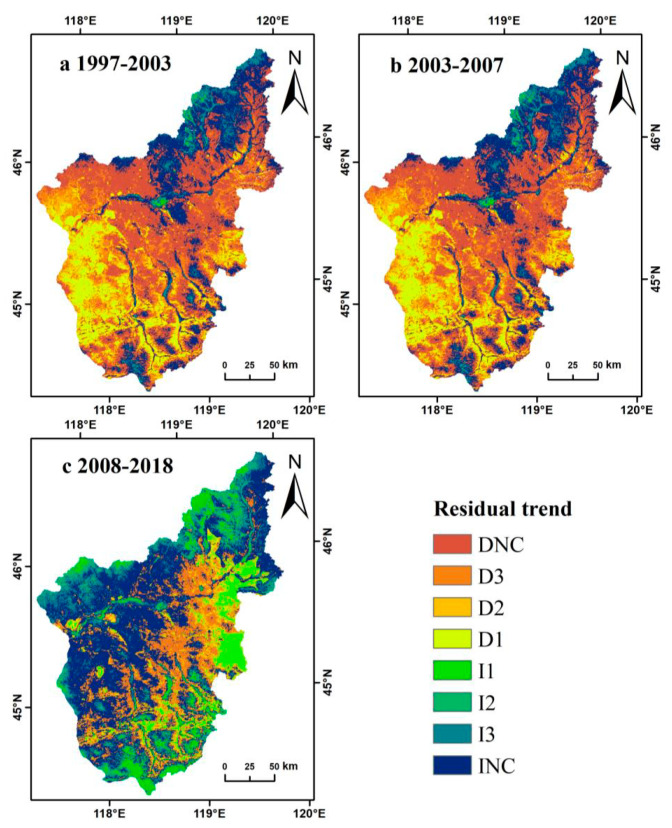
Trends of residuals in time-series.

**Figure 10 ijerph-18-02528-f010:**
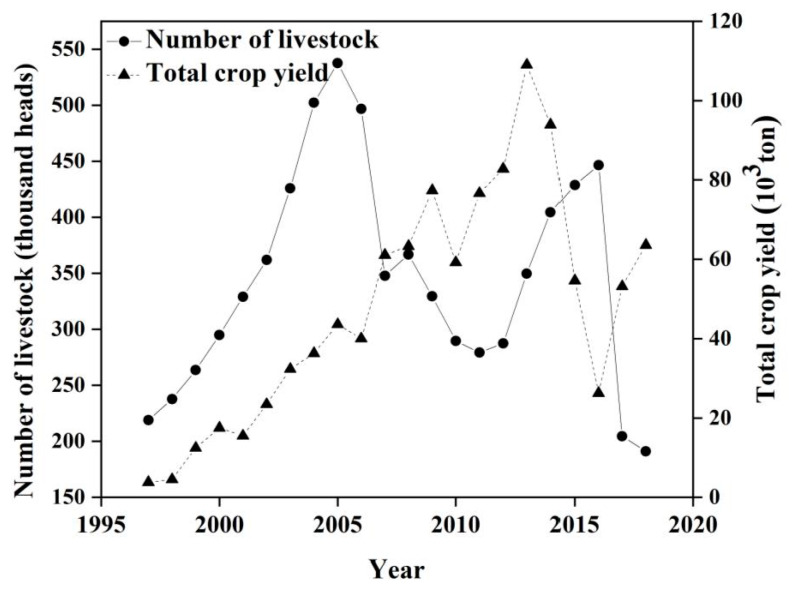
Changes in the number of livestock and crop yield in the Wulagai management area from 1997 to 2018.

**Figure 11 ijerph-18-02528-f011:**
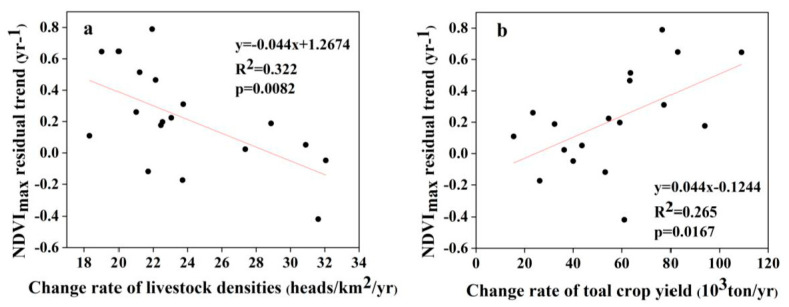
Relationships between NDVI residual trend and change rates: (**a**) for livestock densities from 2001 to 2018; and (**b**) for total crop yields from 2001 to 2018.

**Table 1 ijerph-18-02528-t001:** Correlation coefficient of NDVI with precipitation and temperature in three ecosystems, with significant area (*p* < 0.05) as a percentage of this type.

	Cumulative Rainfall Period	Rainfall April–July	Rainfall April–August	Rainfall June–August	Ln (Rainfall April–July)	Ln (Rainfall April–August)	Ln (Rainfall June–August)	Cumulative Temp January–August	Ln (Cumulative Temp January–August)
Meadow steppe	Correlation coefficient	0.32	0.22	0.22	0.39	0.18	0.20	−0.25	−0.04
	Percentage of *p* < 0.05	27.62	6.84	7.66	49.66	4.72	5.56	11.95	1.22
Typical steppe	Correlation coefficient	0.33	0.27	0.30	0.41	0.28	0.33	−0.21	−0.09
	Percentage of *p* < 0.05	32.36	24.73	32.03	51.65	27.77	36.17	10.56	1.28
Riparian wet meadow	Correlation coefficient	0.25	0.23	0.30	0.33	0.23	0.30	−0.115	−0.025
	Percentage of *p* < 0.05	22.21	16.81	35.41	42.60	16.61	35.23	3.96	1.65

**Table 2 ijerph-18-02528-t002:** Proportions of residual trends for each sub-period in the study area.

Residual	1997–2003 (%)	2003–2007 (%)	2008–2018 (%)
D1, D2, D3	36.31	1.76	15.58
DNC	37.75	16.01	18.37
INC	21.08	58.96	39.08
I1, I2, I3	4.86	23.27	26.97
Total	100.00	100.00	100.00

Note: D1, D2, D3 and DNC denote decreasing trends; I1, I2, I3 and INC denote increasing trends.

**Table 3 ijerph-18-02528-t003:** Coefficients of determination and directions of regression between NDVI residual trend and rate of change in livestock densities and rate of change in total crop yields for 1997–2001 and 2001–2018.

Human Factors	Period	Coefficient of Determination (Direction)
Rate of change in livestock densities	1997–2001	0.009 (−)
	2001–2018	0.044 ** (−)
Rate of change in total crop yield	1997–2001	3.38E-5 (−)
	2001–2018	6.225E-6 * (+)

** *p* < 0.01, * *p* < 0.05.

**Table 4 ijerph-18-02528-t004:** Species composition of three ecosystems.

Ecosystem Type	Years	Community Type	Main Species	Average Plant Height (cm)	Above-Ground Biomass (g/m^2^)
Meadow steppe	1997	*Stipa baicalensis* + *Filifolium sibircum*	*Stipa baicalensis*, *Filifolium sibircum*, *Carex pediformis*, *Artemisia tanacetifolia*, *Leucopoa albida*	18.21 ± 0.5	148.31 ± 3
	2018	*S. baicalensis* + *Carex korshinskyi*	*Stipa baicalensis*, *Carex korshinskyi*, *Sanguisorba officinalis*, *Filifolium sibircum*, *Serratula centauroides*	20.77 ± 0.2	159.02 ± 2
Typical steppe	1997	*S. grandis* + *Leymus chinensis*	*Stipa grandis*, *Leymus chinensis*, *Artemisia frigida*, *Euphorbia fischeriana*, *Scutellaria baicalensis*	14.17 ± 0.2	83.96 ± 5
	2018	*S. grandis* + *S. krylovii* + *L. chinensis*	*Stipa grandis*, *Stipa krylovii*, *Leymus chinensis*, *Euphorbia fischeriana*, *Artemisia frigida*, *Alium ramosm*	12.53 ± 0.3	60.89 ± 3
Riparian wet meadow	1997	*Agrostis alba* + *Potentilla anserina*	*Agrostis alba*, *Potentilla anserina*, *Halerpestes ruthenica*, *Suaeda glauca*, *Carex korshinskyi*	14.66 ± 0.5	196.26 ± 4
	2018	*C. korshinskyi* + *Hemerocallis minor*	*Carex korshinskyi*, *Hemerocallis minor*, *Agrostis alba*, *Potentilla anserina*, *Suaeda glauca*	18.78 ± 0.3	233.16 ± 2

## Data Availability

The data presented in this study are available in the tables of this article. The data presented in this study are available on request from the corresponding author.

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
