# Peer review of "Vegetation Dynamic Assessment by NDVI and Field Observations for Sustainability of China’s Wulagai River Basin"

_ijerph, 2021, doi:10.3390/ijerph18052528_

Round 1
Reviewer 1 Report
Dear authors and editor,
The manuscript reports the observation of vegetation changes under different policy and environmental conditions based on remote data fusion. The study is interesting and may represent a tool for long-term land management evaluation. However, I have some issues with the manuscript, which must be reorganised to improve the reading flow and correct several typos. Moreover, many points must be clarified, as reported in the specific comments. My major concern is about the real reason for vegetation changes, as both climate and management influence its performance. Normalising data could help. It is well known that multispectral time-series have some issues, as NDVI is the response to several vegetation and environment conditions. The authors could reflect on this or explaining how they can discriminate between different options. Here are some examples of NDVI normalisation: https://doi.org/10.1002/ecs2.1919, https://doi.org/10.1002/ecs2.1919
Here are some specific comments:
Line 52: do you mean AVHRR here?
Line 55: greater than what?
Lines 66-67: it’s better to rephrase this sentence
Lines 68-73: this paragraph is not clear, the key differences of the models should be highlighted
Lines 76-77: this sentence is not complete
Line 85: what’s VOD?
Line 90: who’s “he”? You should be more formal; you are referring to “some research”
Lines 123-128: I think you have a third objective, which is to verify if RESTREND method can represent the true succession status of plant communities
Line 168: each species or each plant?
Line 180: what did you use the weather stations data for? To derive data for the current year? Or to validate data collected from the online dataset? As it is not clear when the ground-based experiment was carried out, it’s quite difficult to understand this point
Line 192: residual trend method should be RESTREND, as the acronym has already been introduced
Lines 198-199: add a reference for this filter
Lines 206-209: I really don’t understand this point
Figure 3: what are tm, tn, Tm, Tn and Tp?
Lines 231-232: how can we be sure that degradation is caused by human activities? Do you have any reference for this statement?
Lines 245-246: accumulated temperature of how many days?
Line 253: aren’t they 8 precipitation variables (4 time lags and their logarithmic form)?
Figure 4: this figure is a bit confusing; I don’t catch the comparisons, I suggest reorganising it keeping ESTARFM and MODIS separated. The legend is missing
Line 309: typo, “analyze” is repeated
Line 320: change “better” with a more appropriate word
Figure 7: the R2 of the riparian wet meadow may be similar to the other ones, but the Figure shows that there is not a real correlation. Please discuss this in the Discussion section
Line 357: do you mean significance?
Line 392: Graminaceae?
Table 4: why is there a dot after Stipa?
Line 417: they who?
Lines 418-419: are you sure the changes are related to policy changes? You have shown a similar pattern with precipitation. How can you be sure the changes are due to management policies?
Lines 445-446: check font size
Reviewer 2 Report
The manuscript entitled " Can the vegetation dynamics based on NDVI time series reflect the status of ecosystem? A case study of the Wulagai River Basin" has novel information and has significant importance to development of the studied region, therefore, providing valuable information on further policies development and practices related to the grassland ecosystems. However, the manuscript needs to be revised for English language and adequate its format to a more formal use of English to match requirements for scientific writing. Below, there are some suggestions for authors to consider:
manuscript title: please consider concise it for easier understanding.
line 26: what pasture policies? management of pastures?
line 33: " policies to increase sustainability on grassland ecosystems".
line 34: does communities refer to plant communities on ecosystems? Also, suggest " anthropogenic impacts" instead of "human activity"
line 38: Please revise the first phrase, needs clarification.
line 41: please remove " obvious". Then, I am not sure if the authors are referring to patterns of vegetation, but, interannual change is confusing and might be misleading.
line 44: what rules are you referring to?
line 47: have instead of has
throughout the text: the use of contraction like "can't" and "doesn't" is not common on scientific writing, please consider revising. Also, in several places, please consider changing "fusion" to "merge"
line 82: "Several studies"
line 85: no abbreviations when starting a phrase ("GIMMS")
line 87: please consider using other expression than " It was pointed out"
line 88: instead of "some" consider "Previous studies"
line 92: please consider another word for "fragile"
line 146: please reword the phrase to remove "its"
line 160-162: Please consider: " Fifty-five random sampling sites were determined to perform vegetation evaluation and collection".
line 168 and others: remove "castrated" for "harvested" or something similar
line 167-168: was density measured or the botanical composition based in dry mass?
line 171-173: usually, when several samples are sampled those observations are sampled to obtain an average, not a sum of area like the authors chose to approach on this dataset. If there is a specific reason for the authors to choose this approach, would be beneficial to have a reference
line 185: please clarify
Figures: please consider enlarging figure and increase quality of image to facilitate visualization.
Reviewer 3 Report
In order to clarify that the vegetation dynamics based on NDVI time series can reflect the status of ecosystem, this research tried to perform several analyses such as (1) to construct the NDVI time series of the Wulagai River Basin, China from 1997 to 2018 by integrating GIMMS NDVI and MODIS NDVI, and (2) to identify the driving force for vegetation changes in the Wulagai River Basin.
I can Judge each analysis was performed in due process. However, it seems that this research a little bit concentrated on the method development such as integrating remotely sensed data and do forth , rather than the discussion on the status of ecosystems.
I will recommend the author(s) should revise the research question or change the title.
Also, the Figures, such as Figure4, Figure5 and Figure9, are difficult to understand because of the lack of legends, year of data and do forth. The author(s) should revise figures and tables according to the story of manuscript.
Lastly, the author(s) should remove the blank pages between L334 to L338. The author(s) should submit their manuscript after checking the style of it.
Round 2
Reviewer 1 Report
The quality of the manuscript has been improved, and the manuscript is suitable for publication after minor reviews:
check for repetition in Appendix images.
Best regards
Author Response
请参阅附件。

Reviewer 2 Report
The manuscript entitled " Can the vegetation dynamics based on NDVI time series reflect the status of ecosystem? A case study of the Wulagai River Basin" contributes to with knowledge on this area. Few comments are below:
line 73, 118 - number/date is missing on reference
Figures - please consider enlarging them. It is difficult to visualize the information.
Appendix A - photos are repeated
Reviewer 3 Report
The manuscript has been significantly improved comparing with the original submission, and I can judge it can be almost accepted in present form.
I found several small mistakes or unclear points. Th author(s) should check and revise these points before final acceptance.
L68: "It is an important to solve" --- "It is important to solve"
L200: "55" --- "fifty-five"
L307-311: "To better reflect ...... and produced residuals." ----- This sentence is a little bit difficult to understand the meaning. I hope the author(s) will paraphrase this sentence to other expression.
P17: This blanc page should be removed.
Table1: The author(s) should close up the space between lines of print.
L 472: "Crop area" may be "crop yield".
Appendix Figure A1 and A2:: Subtitles should be added for each photograph for understanding what do these photographs means.
